# Experimental Evaluation of Mode II fracture Properties of *Eucalyptus globulus* L.

**DOI:** 10.3390/ma13030745

**Published:** 2020-02-06

**Authors:** Almudena Majano-Majano, Antonio José Lara-Bocanegra, José Xavier, José Morais

**Affiliations:** 1Department of Building Structures and Physics, ETS Architecture, Universidad Politécnica de Madrid, Avda Juan de Herrera 4, 28040 Madrid, Spain; almudena.majano@upm.es (A.M.-M.); antoniojose.lara@upm.es (A.J.L.-B.); 2UNIDEMI, Department of Mechanical and Industrial Engineering, NOVA School of Science and Technology, Universidade NOVA de Lisboa, 2829-516 Caparica, Portugal; 3School of Science and Technology, CITAB, University of Trás-os-Montes e Alto Douro, 5001-801 Vila Real, Portugal; jmorais@utad.pt

**Keywords:** Eucalyptus globulus, fracture, Mode II, R-curve, cohesive law, ENF, DIC, timber

## Abstract

*Eucalyptus globulus* Labill is a hardwood species of broad growth in temperate climates, which is receiving increasing interest for structural applications due to its high mechanical properties. Knowing the fracture behaviour is crucial to predict, through finite element models, the load carrying capacity of engineering designs with possibility of brittle failures such as elements with holes, notches, or certain types of joints. This behaviour can be adequately modelled on a macroscopic scale by the constitutive cohesive law. A direct identification of the cohesive law of *Eucalyptus globulus* L. in Mode II was performed by combining end-notched flexure (ENF) tests with digital image correlation (DIC) for radial-longitudinal crack propagation system. The critical strain energy release for this fracture mode, which represents the material toughness to crack-growth, was determined by applying the Compliance Based Beam Method (CBBM) as data reduction scheme and resulted in a mean value of 1.54 N/mm.

## 1. Introduction

For a policy of sustainability, engineering applications employing wood and wood-based products are receiving major interest in recent years due mainly to their environmental advantages. Consequently, softwood species such as spruce and pine have been deeply studied because of their wide availability. On the contrary, research concerning hardwood species is scarce. However, the increase in forest area dedicated to temperate hardwoods and the need to increase timber supply in the market are encouraging different European countries to explore the potential of these species for use in construction and, especially, for application as structural elements.

The use of timber and its derivative products for structural purposes requires knowing the response of the material to all possible loading situations, especially those that may produce brittle failures. There are many engineering solutions in timber, such as connections loaded at an angle to the grain, beams with holes, or beams with notches at the supports, which generate perpendicular to the grain and shear stresses that could cause a sudden failure of the element. Regarding the analysis of these situations, an energetic approach, in the framework of fracture mechanics, results very suitable. In fact, various current design codes of timber structures include practical expressions, based on fracture mechanics, for the verification of the load carrying capacity of this type of solutions [1,2]. These expressions have generally been derived from extensive experimental campaigns using timber elements of structural size. Nevertheless, this approach has the drawback of being only valid for the geometric configurations and species under testing; typically, they have been applied to softwoods. Additionally, experimental research can be costly and time-consuming, thus it might not be feasible in some situations.

An alternative to experimental approaches is the computational analysis by means of finite elements models (FEM). They incorporate fundamental mechanical parameters to simulate the material behaviour. Regarding numerical fracture models in wood, it must be noted that fracture behaviour can be affected by nonlinear phenomena such as fibre bridging and micro-cracking along a significant fracture process zone (FPZ) ahead of the crack tip. This behaviour can be adequately modelled on a macroscopic scale by the constitutive cohesive law. This law can be indirectly determined in a recursive way by a load–displacement curve fitting, but the shape of such law must be assumed a priori and therefore significantly affects the fracture results [3]. Another way to directly identify the cohesive law is through the relationship between strain energy release rates (*G*) and crack tip opening displacements (*w*) [4,5], which can be experimentally measured using image-based methods such as digital image correlation technique (DIC) [6,7]. This identification method does not require a priori a cohesive law shape assumption.

The failure modes to be considered in the vast majority of timber engineering applications with the possibility of brittle failures are reduced to Modes I and II (perpendicular tension loading and longitudinal shear loading, respectively). Design situations that may produce other different failure modes are rare. Several FEM investigations in this regard considering cohesive zone modelling and fracture parameters in Modes I and II have confirmed the great utility of this approach to reproduce the fracture process and accurately predict the load carrying capacity of the solutions analysed. Again, most of these works have been performed on softwood species [8,9].

*Eucalyptus globulus* L. is a temperate hardwood with growth in southern Europe, Australia and South America. Together with beech and ash, eucalyptus is one of the three species in Europe with the highest mechanical performance. It is characterised for structural use with an assigned strength class of D40 [10]. Recent research related to the development and characterisation of engineered timber products has highlighted the growing industrial interest in this species [11,12,13]. Likewise, recent works have provided relevant information for the development of FEM numerical models such as the elastic orthotropic characterisation [14] and the determination of the strain energy release rate (*G*_I_) and cohesive law in fracture Mode I [15,16]. However, studies on the fracture behaviour in Mode II of *E. globulus* have not yet been carried out.

This study addressed the direct identification of cohesive law of *Eucalyptus globulus* L. in Mode II obtained by combining end-notched flexure (ENF) tests with digital image correlation. The radial-longitudinal (RL) crack propagation system was analysed. The strain energy release for this fracture mode (*G*_II_) was determined directly from the load–displacement curve by applying the Compliance Based Beam Method (CBBM) as data reduction scheme. It allowed obtaining the resistance curve (*R*-curve) without requirements to monitor the crack length during propagation, which would be a difficult task in such heterogeneous material. The cohesive law was determined by differentiated the *G*_II_–*w*_II_ relationship and reconstructed by means of least-squares regression. The fracture parameters in Mode II obtained represent important information as input parameters for the development of numerical models to study the fracture behaviour of timber structures.

## 2. Materials and Methods

### 2.1. Raw Material

The timber species that was the object of this study was *Eucalyptus globulus* Labill from Galicia, a northwest region of Spain. Ninety-eight kiln-dried boards, classified for structural use in accordance with the Spanish visual grading standard UNE 56546:2013 [17], were firstly tested to select boards of similar mechanical properties. The density (*ρ*) and longitudinal modulus of elasticity (*E*_L_) were determined for every board according to EN 408:2011 [18]. Five boards with similar density (close to the mean value specified at UNE 56546:2013 for eucalyptus, *ρ*_mean_ = 797 kg/m^3^) and similar modulus of elasticity were chosen to prepare the specimens. It is worth noting that the boards were approximately knot-free, which is a characteristic feature of this species. The results from each board (identified with a number) for a reference moisture content of 12% are shown in Table 1, together with mean value, standard deviation (SD) and coefficient of variation (CoV). The same boards were also used in a previous work by the authors to determine the Mode I fracture behaviour of *E. globulus* [16].

The engineering elastic constants of the material are required for the data reduction method followed in the present work, as detailed in Section 2.2. Since it is focused on RL (radial-longitudinal) fracture system (common fracture system in the context of structural application of timber), the required properties in addition to *E*_L_, are the radial modulus of elasticity (*E*_R_) and the shear modulus of elasticity in LR plane (*G*_LR_). The corresponding values were taken from Crespo *et al.* [14], who used Galician *E. globulus* with similar density to the boards in the present study. Mean values of *E*_R_ = 1820 MPa and *G*_LR_ = 1926 MPa were determined from compression tests coupled with a digital image correlation system (DIC) to measure the strain fields.

End-notched flexure (ENF) specimens were prepared from these boards according to the specifications detailed in Section 2.3. The specimens were named using the board reference from which they were obtained. 

### 2.2. Data Reduction: Compliance-Based Beam Method (CBBM)

The cohesive law in Mode II is defined as the relationship between shear stresses, *σ*_II_, and crack tip opening displacements in Mode II (displacements in the direction of the crack propagation associated to the upper and the lower cracked surface), *w*_II_, that is the function: *σ*_II_ = *f*(*w*_II_).

The data reduction method applied to determine this cohesive law in Mode II corresponds to a direct method. It requires differentiating the relationship between the strain energy release rates, *G*_II_, and crack tip opening displacements, *w*_II_, as
(1)$$\sigma_{\mathrm{II}}\;(w_{\mathrm{II}})=\frac{\partial G_{\mathrm{II}}}{\partial w_{\mathrm{II}}}$$
being *G*_II_ accordingly expressed as [4]:(2)$$G_{\mathrm{II}}=\int_{0}^{w_{\mathrm{II}}}\sigma_{\mathrm{II}}({\bar{w}}_{\mathrm{II}})\;d{\bar{w}}_{\mathrm{II}}$$

To directly solve Equation (1), the evolution of *G*_II_ with regard to *w*_II_ in the course of a fracture test must be determined using suitable methodologies. End-notched flexure (ENF) tests (see details in Section 2.3) were performed.

The classical data reduction schemes used to evaluate *G*_II_ are based on beam theory or compliance calibration. Considering the Irwin–Kies equation [19], *G*_II_ is expressed as
(3)$$G_{\mathrm{II}}=\frac{P^{2}}{2B}\frac{dC}{da}$$
with *B* denoting the specimen width, *C* the specimen compliance and *a* the crack length. Therefore, it requires crack-length monitoring during testing [20,21,22,23], which is quite a difficult task in wood considering the toughening mechanisms such as microcracking, crack-branching or fiber-bridging produced at the fracture process zone (FPZ) ahead of the crack tip. Therefore, the classical data reduction methods based on crack length measurements can induce important errors on the *G*_II_ evaluation.

To overcome this limitation, a powerful method based on an equivalent crack length (*a*_eq_) concept is followed instead, which is called *Compliance-based beam method* (CBBM). It makes possible the determination of *G*_II_ by explicitly processing the global load–displacement curve obtained from the test without requirements of crack length monitoring. This method is therefore less sensitive to experimental inaccuracies.

Following CBBM and assuming beam theory with consideration of shear effects, the specimen compliance during crack propagation is written as [24]
(4)$$C=\frac{3a_{\mathrm{eq}}^{3}+2L^{3}}{12E_{f}I}+\frac{3L}{5G_{\mathrm{LR}}A}$$
where *G*_LR_ is the shear modulus of the material; *A* = 2*hB* is the cross-section area; *I* = 8*Bh*^3^/12 is the second moment of area; *a*_eq_ = *a* + Δ*a*_FPZ_ denotes the equivalent crack length, with Δ*a*_FPZ_ the crack length correction accounting for the FPZ effect; and *E_f_* is a corrected flexural modulus which replaces *E*_L_ to take into account the variability in elastic properties of wood. *E_f_* can be estimated from Equation (4) following the next expression:(5)$$E_{f}=\frac{3a_{0}{}^{3}+2L^{3}}{12I}{\left(C_{0}-\frac{3L}{5G_{\mathrm{LR}}A}\right)}^{-1}$$
being *a*_0_ the initial crack length and *C*_0_ the initial compliance of the specimen.

Accordingly, the equivalent crack length during propagation can be determined as follows
(6)aeq=a+ΔaFPZ=[CcorrC0corra03+23(CcorrC0corr−1)L3]13
where
(7)$$C_{\mathrm{corr}}=C-\frac{3L}{5G_{\mathrm{LR}}A};{C_{0}}_{\mathrm{corr}}=C_{0}-\frac{3L}{5G_{\mathrm{LR}}A}$$

Considering the parameters shown in Equations (4)–(7) by applying the CBBM, the general expression to determine the energy release rate in Mode II (Equation (3)) is particularised as
(8)$$G_{\mathrm{II}}=\frac{9P^{2}}{16B^{2}E_{f}h^{3}}{\left[\frac{C_{\mathrm{corr}}}{C_{0_{\mathrm{corr}}}}a_{0}^{3}+\frac{2}{3}\left(\frac{C_{\mathrm{corr}}}{C_{0_{\mathrm{corr}}}}-1\right)L^{3}\right]}^{2/3}$$

*G*_II_ represents de resistance curve (*R*-curve) of the material to the crack growth. As shown above, CBBM does not require crack length monitoring during testing, which evinces the great potential and advantages of the method for this purpose. The only experimental measures needed are the loads and corresponding displacements which are directly recorded in every test. 

The *w*_II_ parameter, needed to be correlated with *G*_II_ (see Equation (1)), was measured by means of image processing based on the digital image correlation (DIC) technique. Speckled images were recorded during the fracture tests, with suitable spatial and temporal resolutions. The numerical image correlation provided the calculation of the displacement fields around the region of interest. Post-processing the displacements at the crack tip allowed an estimation of the crack tip opening displacement during the fracture test. The initial crack tip location *(x_c_^i^, y_c_^i^)* was firstly identified in a reference image before crack propagation. At that coordinate location, the *uy(:,y_c_^i^)* displacement profile was evaluated in a direction perpendicular to the crack growth. The crack tip opening displacement was then evaluated as the major different along the profile, corresponding to the real jump in terms of displacements at the two cracked surfaces. This approach allowed systematically considering any path deviation of the crack propagation during the test.

The *G*_II_–*w*_II_ curve is finally differentiated to obtain the cohesive law in Mode II. In the process, a continuous approximation function (logistic function) was used to fit the *G*_II_–*w*_II_ experimental data according to Equation (9).
(9)$$G_{\mathrm{II}}=\frac{A_{1}-A_{2}}{1+{\left(w_{II}/w_{II,0}\right)}^{p}}+A_{2}$$
where *A*_1_, *A*_2_, *p* and *w*_II,0_ constants were obtained by least-square regression analysis [25]. Even if this function does not have a particular physical meaning, it is used instead to provide filtering and analytical differentiation in the reconstruction of the cohesive law [26]. From such relationship, the *A*_2_ parameter should provide an estimation of the critical strain release in Mode II, as
(10)$$A_{2}=\underset{w_{II}\to \infty }{\lim}G_{II}=G_{IIc}$$

### 2.3. End-notched Flexure Tests

End-notched flexure (ENF) tests were proposed in this work. They consist of pre-cracked beam specimens which are loaded in a three-point bending configuration. Ten ENF specimens were cut from dried boards of *E. globulus*. The nominal dimensions were *L*_1_ mm × 2*h* mm × *B* mm (500 mm × 20 mm × 20 mm), as schematically shown in Figure 1. All specimens were oriented along the RL crack propagation system (Radial loading direction and Longitudinal crack propagation direction). A mid-height sharp pre-cracked surface of around 162 mm in length was initially performed by a band saw of 1 mm thickness and lengthened around 1–2 mm afterwards using an impacted blade. The actual value of initial crack length (*a*_0_) was more clearly measured after testing once the specimen was broken and thus divided in two parts.

Prior to testing, the specimens were stabilised at laboratory conditions of 20 °C and 65% relative humidity.

Fracture tests were performed using an INSTRON 1125 universal testing machine (Instron, Barcelona, Spain) with a load cell of 5 kN maximum capacity and 200 N/V gain. The specimens were placed on two cylindrical rollers with 2*L* = 460 mm span and the load was applied at mid-span using a cylindrical actuator to minimise indentation. To reduce frictional forces during testing, a Teflon film was placed along the pre-cracked surfaces of the specimen. The tests were carried out under displacement control with a cross-head velocity of 3 mm/min.

A non-contact optical system, ARAMIS 2D (GOM mbH, Braunschweig, Germany) [27], was coupled with the test device in order to measure crack tip opening displacements at the specimens during fracture tests. This technique applies the principles of digital image correlation (DIC) and makes possible full-field measurements, which lead to more robust results in comparison with conventional methods. ARAMIS 2D system is composed by an eight-bit charge-coupled device (CCD) camera with a telecentric lens, and two cold light sources to illuminate the specimens (Figure 2, left). Details of the DIC setting parameters are shown in Table 2. A subtle speckle pattern of black ink points on a white matte surface was applied at the crack tip area of every specimen using an airbrush IWATA, model CM-B (Anesta Iwata Iberica SL, Barcelona, Spain) in order to have proper granulometry contrast and isotropy at the magnification scale (Figure 2, right). Both the speckle pattern quality and the DIC setting parameters must be property selected to achieve a suitable compromise in terms of spatial resolution and accuracy [28,29,30]. In particular, a spatial resolution of 0.270 mm was decided in this work for a displacement accuracy around 1–2 × 10^−2^ pixel (0.18–0.36 μm^2^). The accuracy in displacements was estimated experimentally by statistically analysing noisy fields obtained by processing images recorded for rigid-body translation tests. *P–δ* data were recorded in all tests with an acquisition rate of 5 Hz. The DIC images acquisition was chosen as 1 Hz frequency. Crack tip opening displacements in Mode II (*w*_II_) were determined by post-processing the displacements monitored by DIC, as specified in Section 2.2.

## 3. Results and Discussion

### 3.1. Resistance Curve

The load–displacement curves resulted from the ENF tests are shown in Figure 3. The curves show quite consistent and similar behaviour considering inherent variability of this heterogeneous material. The effect of such heterogeneity may also explain the uneven and gradual decrease of load after the peak [31]. The non-linear behaviour observed before peak load is related to the development of the FPZ ahead of the crack tip. 

The maximum loads reached in the fracture tests are summarised in Table 3. Even if they show some scatter typical of the material, all values remain approximately in a similar range with an average value of 706 N.

From every *P–δ* curve, the initial compliance *C*_0_ was calculated using MATLAB^®^ software as the value that gives the maximum *R*^2^. An example of the compliance–*R*^2^ relationship and the corresponding *P–δ* curve for a particular ENF specimen is illustrated in Figure 4. The *C*_0_ results from all tests are included in Table 3.

The *R*-curves were then evaluated from the experimental *P–δ* curves by applying the CBBM data reduction method detailed in Section 2.2. They are shown in Figure 5 as relationship between equivalent crack length and energy release rate. These curves are characterised by an initial rising domain meaning that the FPZ is developing, and a horizontal asymptote afterwards which represents the material toughness to crack-growth, the so-called strain energy release rate (*G*_IIc_). This is one of the most important parameters that define the material fracture behaviour in Mode II.

The *R*-curves scatter is within the expected range for wood and may be attributed to the variability of the material microstructure at the initial crack tip [26]. However, most of the specimens showed a *plateau* for a given crack extent. Therefore, *G*_IIc_ could be derived as a mean value over data covering the *plateau* domain. These values are included in Table 3 for each specimen as well as the strain energy release rate corresponding to the maximum load, *G*_II,Pmax_.

There is no good correlation (R^2^ = 0.10) between flexural modulus obtained by the ENF tests (also included in Table 3) and the *G*_IIc_ values. The mean *G*_IIc_ value resulted in 1.54 N/mm. This value is twice the *G*_Ic_ value (0.77 N/mm) obtained from DCB tests under Mode I using also *Eucalyptus globulus* of the same quality in previous work by the authors [16]. *E. globulus* also shows higher *G*_IIc_ values than other species. For instance, *Pinus pinaster* subjected to the same ENF tests and applying CBBM as data reduction method led to an average *G*_IIc_ of 1.15 N/mm [26]. *G*_IIc_ values ranging from 0.39 to 0.55 N/mm depending of the analysis method were obtained for Sitka spruce from ENF tests [32]. Studies on Western hemlock derived in *G*_IIc_ values in the range of approximately 0.20–0.55 N/mm depending on the initial crack length in ENF specimens [33]. 

As can be seen in the table, the *G*_II,Pmax_ values resulted similar to those of *G*_IIc_. Therefore, the first one could be assumed as critical strain energy release rate value in Mode II in a practical way or in cases where *R*-curve does not show a clear horizontal asymptote. It would mean that a critical energy would be already consumed at maximum load on the onset of a fully developed FPZ just before steady-stage crack propagation [22,24].

### 3.2. Cohesive Law

The cohesive laws in Mode II of the *E. globulus* specimens were obtained from the relationship between *G*_II_ and transverse crack tip displacements in Mode II (*w*_II_) measured by DIC during testing. Figure 6 shows the evolution of transverse crack tip displacements in Mode II (sliding) and also normal crack tip displacements corresponding to a Mode I during testing of an ENF specimen. As derived from the results, crack tip displacements were found to be predominant in transverse direction following Mode II for a representative extension of the crack propagation. However, some Mode I crack opening can eventually be observed in most of the specimens mainly at the end of the ENF test because there may not be a perfect match between the specimen neutral plane and the initial crack surface.

Typical macroscopic fracture behaviour of the ENF specimens before and after crack propagation can be observed in Figure 7 (left and right, respectively), which clearly evidence the predominant sliding Mode II of fracture mentioned before. 

*G*_II_ values were finally correlated with *w*_II_, as shown in Figure 8 (left). The experimental results were filtering by least-square regression using a logistic function. Figure 8 (right) illustrates both experimental and logistic approximation data for a ENF specimen. As can be seen, a reasonably accurate fitting between the curves is displayed.

The cohesive laws in Mode II expressed by *σ*_II_–*w*_II_ relationship were obtained by analytical differentiation of the *G*_II_–*w*_II_ curves. The results for each specimen are shown in Figure 9. The parameters describing the logistic function of every cohesive law (*A*_1_, *A*_2_, *p* and *w*_II0_ in Equation (9)), the area circumscribed by the cohesive laws (*G*_law,II_), the maximum stress (*σ*_IIu_) and the relative displacements in Mode II corresponding to maximum stress (*w*_IIu_) are summarised in Table 4. 

The mean value of *A*_2_ parameter provides an estimation of the critical strain energy release rate, *G*_IIc_. In the present case, a value of *G*_Iic_ = 1.63 N/mm was obtained, which is just 5.6% higher than the *G*_IIc_ value determined from the *R*-curves (1.54 N/mm, see Table 3) even considering the scatter exhibited by the results. A similar pattern in terms of dispersion was observed for other species in a related work on *Pinus pinaster* [26].

The mean experimental cohesive law in Mode II for *Eucalyptus globulus* L. can be finally built from the mean parameters presented in Table 4. It is highlighted by a bold curve in Figure 9. This cohesive law could be implemented in finite element cohesive zone models to simulate the development of the FPZ and crack growth, and thus study the fracture behaviour of timber structures with possibility of brittle failures with Mode II component.

## 4. Conclusions

The fracture properties in Mode II of *Eucalyptus globulus* L. in RL crack propagation system were experimentally determined by coupling end-notched flexure (ENF) tests with digital image correlation. 

The resistance curves (*R*-curves) of the material are presented, which were derived by applying the CBRM considering an equivalent crack length and thus overcoming inherent difficulties of measuring the actual crack length during propagation of a highly heterogeneous and orthotropic material.

The mean value of critical strain energy release rate in Mode II (*G*_IIc_) resulted in 1.54 N/mm, which is twice the mean value reported for Mode I in previous work by the authors on the same wood species. The *G*_IIc_ value obtained for eucalyptus was also greater than that of other species reported in the literature.

The fracture cohesive laws of eucalyptus, expressed as the relationship between stresses and relative displacements in Mode II, measured by means of digital image correlation, are shown. The laws definition could be accurately implemented in finite element models to predict the crack growth along a fracture process zone.

The fracture properties in Mode II presented make it possible to quantify the fracture behaviour of this potential species in timber engineering situations involving this type of failure.

## Figures and Tables

**Figure 1 materials-13-00745-f001:**
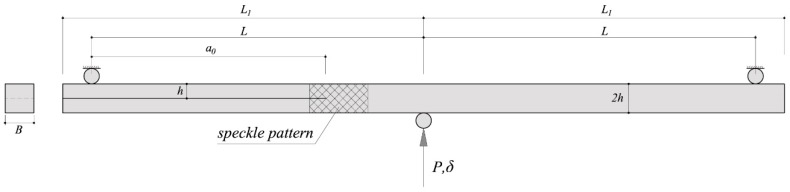
ENF test set-up.

**Figure 2 materials-13-00745-f002:**
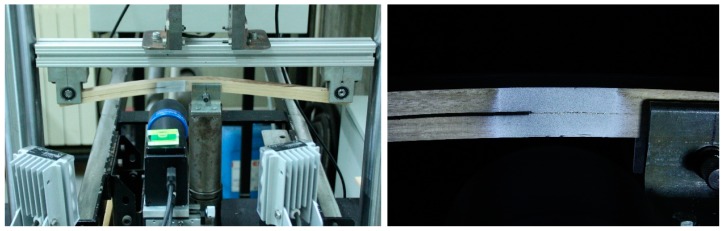
ENF test set-up coupled with DIC (**left**); and speckle pattern detail (**right**).

**Figure 3 materials-13-00745-f003:**
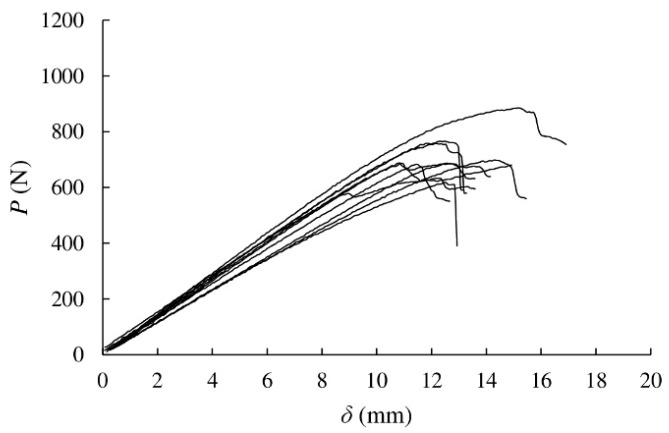
Load–displacement curves obtained from 10 eucalyptus specimens subjected to ENF tests.

**Figure 4 materials-13-00745-f004:**
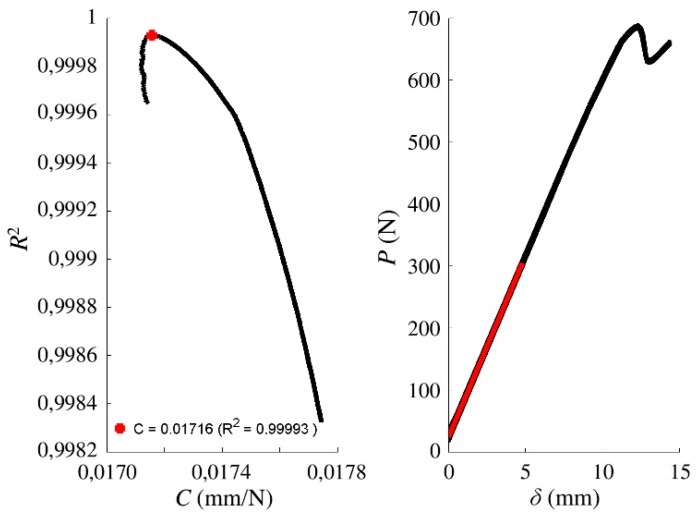
Example of initial compliance determination for a load–displacement curve from an ENF test (specimen “176-3”): (Left) compliance–*R*^2^ relationship (red point indicates the compliance with maximum *R*^2^); and (right) load–displacement curve with the compliance highlighted in red.

**Figure 5 materials-13-00745-f005:**
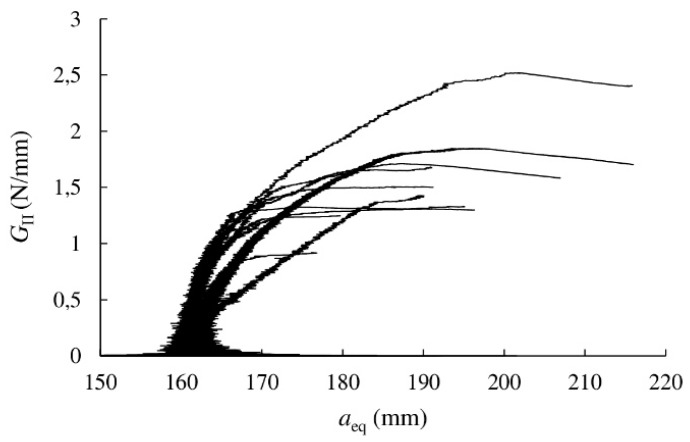
*R-*curves obtained from 10 specimens of *E. globulus* subjected to ENF tests.

**Figure 6 materials-13-00745-f006:**
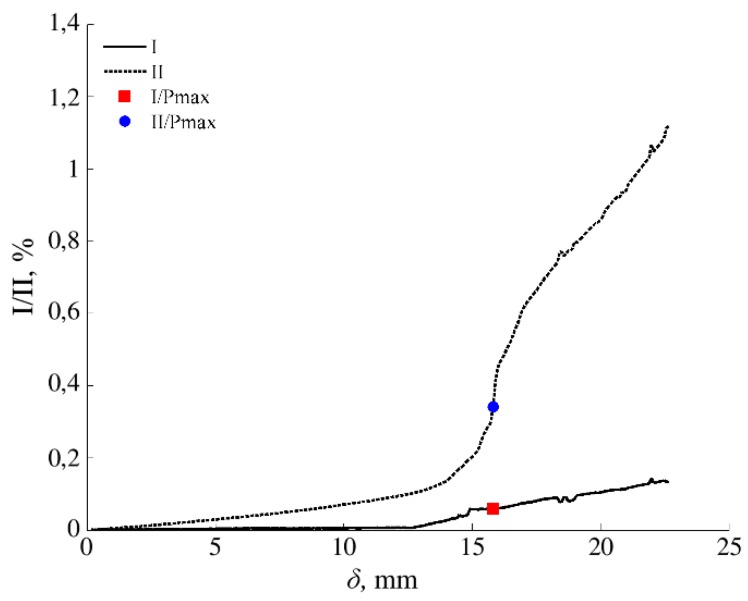
Normal (I) and transverse (II) crack tip opening displacements in relation to crack extension measured by DIC during an ENF test (specimen “144-1”).

**Figure 7 materials-13-00745-f007:**
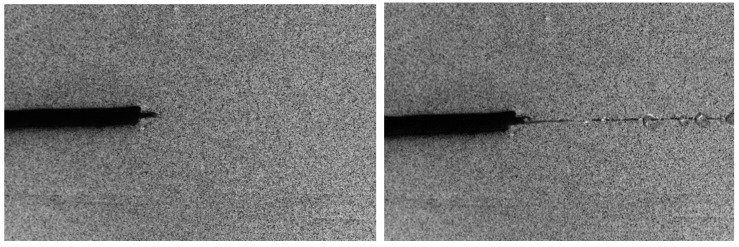
Macroscopic images of ENF 192-1 specimen before (**left**) and after (**right**) crack propagation.

**Figure 8 materials-13-00745-f008:**
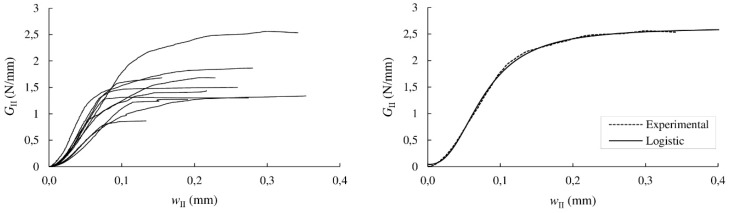
Experimental *G*_II_–*w*_II_ curves (**left**); and experimental *G*_II_–*w*_II_ curve of the ENF 144-1 specimen and fitting with the logistic function (**right**).

**Figure 9 materials-13-00745-f009:**
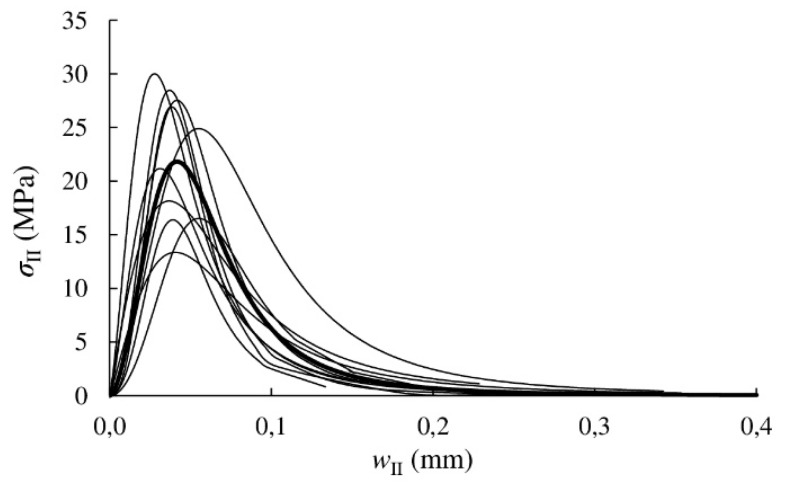
Cohesive laws. Mean cohesive law is highlighted in bold.

**Table 1 materials-13-00745-t001:** Density and elastic modulus of elasticity of *E. globulus* boards.

Board ref.	*ρ* (kg/m^3^)	*E*_L_ (MPa)
144	765	19234
161	867	19658
176	779	19359
189	748	19114
192	815	20612
mean	795	19595
SD	47	603
CoV (%)	6.0	3.1

**Table 2 materials-13-00745-t002:** DIC parameters.

	Settings
CCD camera	
Model	Baumer Optronic FWX20 (8 bits, 1624 × 1236 pixels, 4.4 μm/pixel)
Shutter time	0.7 ms
Acquisition frequency	1 Hz
Lens	
Model	Opto Engineering Telecentric lens TC 23 36
Magnification	0.243 ± 3%
Field of view (1/1.8”)	29.3 × 22.1 mm^2^
Working distance	103.5 ± 3 mm
Working F-number	*f*/8
Field depth	11 mm
Conversion factor	0.018 mm/pixel
Lighting	Raylux 25 white-light LED
DIC measurements	
Subset size	15 × 15 pixel^2^ (0.270 × 0.270 mm^2^)
Subset step	13 × 13 pixel^2^ (0.234 × 0.234 mm^2^)
Resolution	1–2 × 10^−2^ pixel (0.18 × 0.36 μm^2^)

**Table 3 materials-13-00745-t003:** Flexural modulus (*E_f_*), maximum load (*P*_max_), initial compliance (*C*_0_), strain energy release rate at maximum load (*G*_II,Pmax_) and critical strain energy release rate (*G*_IIc_) in Mode II by CBBM.

Specimen Ref.	*E_f_* (MPa)	*P*_max_ (N)	*C*_0_ (mm/N)	*G*_II,Pmax_ (N/mm)	*G*_IIc_ (N/mm)
144-1	17,576	888.32	0.014	2.42	2.50
144-2	14,440	701.23	0.018	1.79	1.84
144-3	14,180	618.65	0.017	1.37	1.38
161-2	16,346	767.78	0.015	1.63	1.70
176-1	14,703	683.96	0.016	1.29	1.32
176-3	14,286	688.20	0.017	1.46	1.50
189-2	16,380	690.00	0.015	1.22	1.24
189-3	15,837	581.88	0.015	0.89	0.90
192-1	17,032	684.18	0.015	1.24	1.28
192-2	16,714	761.13	0.015	1.51	1.70
Mean	15,749	706.53	0.016	1.65	1.54
SD	1251	84.78	0.001	0.41	0.44
CoV (%)	8	12	9	25	28

**Table 4 materials-13-00745-t004:** Logistic function parameters (*A*_1_, *A*_2_, *p* and *w*_II0_), maximum stress (*σ*_IIu_) and relative displacement (*w*_IIu_), as determined by CBBM equations.

Ref.	*A*_1_ (N/mm)	*A*_2_ (N/mm)	*p*	*w*_II0_ (mm)	*G*_law,II_ (N/mm)	*σ*_IIu_ (MPa)	*w*_IIu_ (mm)
144-1	0.042	2.62	2.52	0.077	2.55	24.89	0.055
144-2	0.017	1.82	2.28	0.042	1.78	30.01	0.028
144-3	0.007	1.47	2.25	0.048	1.42	21.17	0.031
161-2	0.027	1.80	2.93	0.053	1.75	27.53	0.042
176-1	0.035	1.41	3.30	0.046	1.42	26.92	0.038
176-3	0.033	1.62	3.00	0.047	1.61	28.46	0.037
189-2	0.035	1.39	3.01	0.069	1.33	16.51	0.055
189-3	0.017	0.93	3.11	0.048	0.92	16.39	0.039
192-1	0.001	1.39	2.05	0.068	1.35	13.36	0.041
192-2	0.002	1.82	1.97	0.065	1.76	18.14	0.037
Mean	0.022	1.63	2.64	0.056	1.59	22.34	0.040
SD	0.015	0.44	0.48	0.012	0.43	5.96	0.008
CoV (%)	69	27	18	22	27	27	22

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
