# Peer review of "Experimental Evaluation of Mode II fracture Properties of Eucalyptus globulus L."

_materials, 2020, doi:10.3390/ma13030745_

Round 1

Reviewer 1 Report

In the manuscript entitled “Experimental evaluation of Mode II fracture 2 properties of Eucalyptus globulus L.” the authors described a direct identification of the cohesive law of hardwood species in mode II using imaging methods. I have several comments and concerns regarding data representation and analysis.

Page 2, line 88. Please clearly state in the text how many trees were used to prepare kiln-dried boards. Page 7, Figure 3 legend. “P-δ curves from ENF tests” please provide an appropriate figure description including statistics. Page 7, Figure 4. Please provide an appropriate description for the figure indicating that the graph is shown on the left and what graph is shown on the right. It is not enough to mention it in the main text, figure legends should be self-explanatory with no need to search the main text to understand its content. Provide statistics for the figure plots, if the plot is representative how many measurements were taken to select the representative graph. Page 8. Provide statistics for Figure 5 in the figure legend. Page 8 line 240, is “not” supposed to be “no”? Page 8, line 240-241, what does it mean “a clear correlation”? Please perform appropriate mathematical analyses to quantify correlation. There is always a correlation that can be quantified. Page 9, Figure 6. How was a representative ENF test determined? Table 3 and Table 4. It is important to understand if the specimens with the same first three digits coming from the same tree of different since the authors average these values thus overrepresenting one original specimen over others. Page 11, line 301, please provide some qualitative description for the obtained value, is it high or is it low in comparison to other hard and softwoods.

Author Response

Response to Reviewer 1 Comments

In the manuscript entitled “Experimental evaluation of Mode II fracture 2 properties of Eucalyptus globulus L.” the authors described a direct identification of the cohesive law of hardwood species in mode II using imaging methods. I have several comments and concerns regarding data representation and analysis.

The authors would like to thank the reviewers for the valuable comments provided to the original version of the manuscript. Accordingly, all questions have been addressed and taken into account with a view to improve the manuscript. A revised version of the paper is prepared, in which carefully considerations of all queries mentioned in the reviewers' comments were included. Changes are mark-up in the new version of the paper. Please, find enclosed a point-by-point response to the reviewer’s comments.

Point 1: Page 2, line 88. Please clearly state in the text how many trees were used to prepare kiln-dried boards.

Response 1: Boards of eucalyptus species used to extract samples for this study were selected according to the procedure clarified in the revised manuscript (no further information was available concerning the number of trees used to obtain the boards):

“98 kiln-dried boards, classified for structural use in accordance with the Spanish visual grading standard UNE 56546:2013 [17], were firstly tested in order to select boards of similar mechanical properties. The density (ρ) and longitudinal modulus of elasticity (EL) were determined for every board according to EN 408:2011 [18]. Five boards with similar density (close to the mean value specified at UNE 56546:2013 for eucalyptus, ρmean=797 kg/m3) and similar modulus of elasticity were chosen to prepare the specimens.”

Point 2: Page 7, Figure 3 legend. “P-δ curves from ENF tests” please provide an appropriate figure description including statistics.

Response 2: The description of Figure 3 caption has been modified for a better understanding. The global P-d curves present the overall behaviour of the tested ENF specimens. Further analysis in terms of both the resistance-curve and cohesive laws will allow for a better interpretation and statistical study.

Point 3: Page 7, Figure 4. Please provide an appropriate description for the figure indicating that the graph is shown on the left and what graph is shown on the right. It is not enough to mention it in the main text, figure legends should be self-explanatory with no need to search the main text to understand its content. Provide statistics for the figure plots, if the plot is representative how many measurements were taken to select the representative graph.

Response 3: The legend has been modified in order to be self-explanatory including the description of both figures (left and right) according to your suggestion.

Statistics are shown at the figure plot (R2).

The plot is an example of the procedure followed to determine the initial compliance using a specific specimen (“176-3”). It is not intended to be a representative statistical curve.

The word “representative” has been deleted from the legend and text to avoid possible misunderstandings. Thank you.

Point 4: Page 8. Provide statistics for Figure 5 in the figure legend.

Response 4: Accordingly Figure 5 legend has been modified and clarified in the revised manuscript.

Point 5: Page 8 line 240, is “not” supposed to be “no”?

Response 5: Right. It has been corrected. Thank you.

Point 6: Page 8, line 240-241, what does it mean “a clear correlation”? Please perform appropriate mathematical analyses to quantify correlation. There is always a correlation that can be quantified.

Response 6: The correlation has been now quantified and added in the text (R2=0.10). Thank you.

Point 7: Page 9, Figure 6. How was a representative ENF test determined? Table 3 and Table 4. It is important to understand if the specimens with the same first three digits coming from the same tree of different since the authors average these values thus overrepresenting one original specimen over others.

Response 7: Figure 6 is just a plot example coming from a particular specimen (“144-1”). It is not intended to be representative statistical data. Again, the word “representative” has been deleted and the legend has been modified for a better understanding.

Specimens with the same first three digits come from the same board (it has been now clarified in the text, Line 111). The references of the corresponding boards are shown in Table 1.

Your comment about the average of the values is of great interest. As previously mentioned, boards with similar mechanical properties (ρ and E) were selected for this study. However, the results indicate that there is no correlation between these properties and GIIc (R2=0.0001 for E-GIIc and R2=0.028 for ρ-GIIc). For this reason and due to the lack of other studies that somehow relate the origin of the specimen (board or tree) with the fracture properties, it was decided to calculate the mean value regardless of that origin. Of course, in-depth analysis of the possible relationship between origin and fracture properties results of great interest but this is beyond the scope of this paper.

Point 8: Page 11, line 301, please provide some qualitative description for the obtained value, is it high or is it low in comparison to other hard and softwoods.

Response 8: Qualitative description has been added in Conclusion section according to your suggestion. In addition, a comparison of the GIIc values obtained for eucalyptus and for other species from literature (using ENF tests) have in included in Results section (line 250…).  

Reviewer 2 Report

The wood is useful renewable resources and materials. Hardwood species is fairly important for construction industry.

This manuscript concerns the mechanical properties of the hardwood Eucalyptus globulus Labill that can be used for load carrying of engineering structure.

Even though this study is adequate for the engineering application, this study is not so novel in the fracturing mechanics.

In a word, this study can be accepted for publication because this is MDPI-style.

Author Response

Response to Reviewer 2 Comments

The wood is useful renewable resources and materials. Hardwood species is fairly important for construction industry.

This manuscript concerns the mechanical properties of the hardwood Eucalyptus globulus Labill that can be used for load carrying of engineering structure.

Even though this study is adequate for the engineering application, this study is not so novel in the fracturing mechanics.

In a word, this study can be accepted for publication because this is MDPI-style.

Response 1: The authors would like to thank the review comments. The methodology indeed it not novel in the sense that the cohesive identification approach described and used in this research work was already validated or used on previous work. Nevertheless, this is the first time that this fracture characterisation procedure is used the Hardwood species of Eucalyptus globulus Labill. This study thus provides a repeatability approach with new data for this species, which can be of interest for engineering application.

Reviewer 3 Report

Dear Editor and Authors,

I have read the article entitled „Experimental evaluation of Mode II fracture properties of Eucalyptus globulus L.".

This is a straightforward study. It appears that the authors brought new knowledge on the fracture behaviour in mode II of E. globulus which has not yet been carried out. This work addresses the direct identification of cohesive law of Eucalyptus globulus L. in mode II when obtained by combining end-notched flexure (ENF) tests with digital image correlation. Moreover the work focuses on a species which is undergoing increasing interest for structural applications due to its high mechanical properties.  

Results of the work represent the input parameters for the development of the numerical models to study the fracture behaviour of timber structures.

The subject of the study falls within the scope of the Journal. The title and abstract reflect the content. The introduction initiates the reader on the subject and a well-selected list of references was used. Nine figures and four tables are compiled and all of them are mentioned within the text.  Comparisons with the results of previous works and explanations are given for a better understanding of the subject.

The paper is ready for publication. I have no observations.

I hope my revision was of help.

Author Response

Response to Reviewer 3 Comments

Dear Editor and Authors,

I have read the article entitled „Experimental evaluation of Mode II fracture properties of Eucalyptus globulus L.".

This is a straightforward study. It appears that the authors brought new knowledge on the fracture behaviour in mode II of E. globulus which has not yet been carried out. This work addresses the direct identification of cohesive law of Eucalyptus globulus L. in mode II when obtained by combining end-notched flexure (ENF) tests with digital image correlation. Moreover the work focuses on a species which is undergoing increasing interest for structural applications due to its high mechanical properties.

Results of the work represent the input parameters for the development of the numerical models to study the fracture behaviour of timber structures.

The subject of the study falls within the scope of the Journal. The title and abstract reflect the content. The introduction initiates the reader on the subject and a well-selected list of references was used. Nine figures and four tables are compiled and all of them are mentioned within the text.  Comparisons with the results of previous works and explanations are given for a better understanding of the subject.

The paper is ready for publication. I have no observations.

I hope my revision was of help.

Response 1: The authors would like to thank the reviewer for the careful reading and encouraging words.